# MS-Based Approaches Enable the Structural Characterization of Transcription Factor/DNA Response Element Complex

**DOI:** 10.3390/biom9100535

**Published:** 2019-09-26

**Authors:** Lukáš Slavata, Josef Chmelík, Daniel Kavan, Růžena Filandrová, Jan Fiala, Michal Rosůlek, Hynek Mrázek, Zdeněk Kukačka, Karel Vališ, Petr Man, Michael Miller, William McIntyre, Daniele Fabris, Petr Novák

**Affiliations:** 1Institute of Microbiology, The Czech Academy of Sciences, 14220 Prague, Czech Republic; lukas.slavata@gmail.com (L.S.); chmelik@biomed.cas.cz (J.C.); kuan@maradan.org (D.K.); ruzena.liskova@gmail.com (R.F.); FiJi.cs@seznam.cz (J.F.); rosulek.michal@gmail.com (M.R.); hynek.mrazek@gmail.com (H.M.); zdenek.kukacka@biomed.cas.cz (Z.K.); karel.valis@biomed.cas.cz (K.V.); pman@biomed.cas.cz (P.M.); 2Faculty of Science, Charles University, 12843 Prague, Czech Republic; 3RNA Institute, University at Albany, State University of New York, Albany, NY 12222, USA; mrmiller@albany.edu (M.M.); wdmcintyre@albany.edu (W.M.); dfabris@albany.edu (D.F.)

**Keywords:** transcription factor, protein, DNA, protein-nucleic acid cross-linking, cross-linking, transplatin, trans-dichlorodiamineplatinum(II), hydrogen-deuterium exchange, FOXO4, molecular modeling

## Abstract

The limited information available on the structure of complexes involving transcription factors and cognate DNA response elements represents a major obstacle in the quest to understand their mechanism of action at the molecular level. We implemented a concerted structural proteomics approach, which combined hydrogen-deuterium exchange (HDX), quantitative protein-protein and protein-nucleic acid cross-linking (XL), and homology analysis, to model the structure of the complex between the full-length DNA binding domain (DBD) of Forkhead box protein O4 (FOXO4) and its DNA binding element (DBE). The results confirmed that FOXO4-DBD assumes the characteristic forkhead topology shared by these types of transcription factors, but its binding mode differs significantly from those of other members of the family. The results showed that the binding interaction stabilized regions that were rather flexible and disordered in the unbound form. Surprisingly, the conformational effects were not limited only to the interface between bound components, but extended also to distal regions that may be essential to recruiting additional factors to the transcription machinery. In addition to providing valuable new insights into the binding mechanism, this project provided an excellent evaluation of the merits of structural proteomics approaches in the investigation of systems that are not directly amenable to traditional high-resolution techniques.

## 1. Introduction

A key role in the regulatory of expression machinery is covered by transcription factors (TFs), proteins that recognize target DNA sequences called response elements and establish specific interactions with additional factors to activate or inhibit the transcription process [1,2,3,4]. The species involved in the process have been unambiguously identified [5,6], but significant information is still lacking on the effects of structure/dynamics on specific recognition and mechanism of action. The Protein Data Bank contains the high-resolution structures of at least 483 TFs from different species [7], which include less than 10% of all predicted human TFs [6]. Of such structures, only one third also include the cognate DNA response element, and only one fifth are available in both bound and unbound states (Figure 1). Due to the size and complexity of such systems, most solved structures do not include the entire TF sequence, but consist almost exclusively of the DNA binding domain (DBD). This fact reflects the modular organization of TFs, which includes discrete domains acting in rather independent manner [8,9]. While DBDs tend to be highly structured, other regions responsible for either modulating transcription activity, or supporting facultative ligand interactions, are rather flexible and assume well-defined conformations only upon binding to the intended factor [10]. The unstructured nature of these regions poses many challenges to conventional high-resolution approaches, which require adequate conformational stability and homogeneity. In most cases, the natural interactions established in vivo, which are responsible for stabilizing well-defined functional conformations, cannot be properly replicated in vitro. These challenges explain the chronic lack of comprehensive information on full-fledged TF structures, which still hampers the elucidation of their mechanism of action at the molecular level.

Powered by the development of new experimental strategies and mass spectrometric (MS) instrumentation, structural proteomics has rapidly become an essential approach for gathering valuable structural information for species that are not directly amenable to conventional high-resolution techniques [11,12,13,14]. In addition to hydrogen-deuterium exchange (HDX) [15], a combination of chemical and biochemical techniques broadly known as MS3D [16,17,18] have been effectively utilized to identify the regions of contact between bound biomolecules and reveal their mutual spatial organization. In HDX experiments, the exchange rate of backbone amide hydrogens, which is affected by solvent accessibility and possible involvement in hydrogen bonding, can be directly determined by MS analysis. This technique has been broadly employed to study conformational changes [19,20], protein folding [21] and protein-protein interactions [22,23], as well as nucleic acid-protein complexes [24,25,26,27,28,29]. Among the MS3D techniques, chemical and photo-activated cross-linking (XL) are employed to generate stable covalent bridges between contiguous functional groups, which can reveal their mutual placement in the targeted assembly [13,16]. A variety of bifunctional reagents with different spacing between reactive groups have been developed to determine the distance between susceptible residues. In this way, the sequence position of cross-linked residues and the length of the respective cross-linker provide valid constraints for building accurate molecular models through established computational methods [17,18,30,31,32]. The excellent versatility of these approaches has prompted the development of reagents capable of targeting functional groups present on protein, as well as nucleic acid substrates [33,34,35,36,37,38,39]. Over time, capture tags and isotopic labels have been included in the cross-linker design to facilitate the isolation and analysis of cross-linked products [40,41]. Isotope-labelling, in particular, has provided a valuable tool for highlighting the presence of different conformational states and quantifying their partitioning on the basis of cross-linking probability [13].

In this study, we evaluated the concerted application of complementary structural proteomics techniques to overcome the challenges posed by full-fledged complexes between TFs and respective DNA response elements. We selected a model system consisting of the human transcription factor FOXO4 and its cognate Daf-16 family member-binding element (DBE) [42]. FOXO4 is part of the “O” subfamily of the forkhead box (FOX) class of transcription factors [42,43,44]. The DBD regions of this class are characterized by winged helix structures, which comprise by approximately 100 amino acids folded into helix-turn-helix motifs and β-sheet-bordered loops that make them resemble butterfly wings [45]. The first high-resolution structure of FOXO4-DBD, which was obtained by NMR spectroscopy, confirmed the presence of a typical forkhead, winged helix fold [46]. At the same time, however, the report indicated that the N- and C- terminal regions of the DBD displayed chemical shifts consistent with highly flexible, disordered structures. The more recent identification of consensus sequences for the FOXO family [47] enabled the crystallization of a complex comprising a selected DBE duplex and a FOXO4-DBD construct that lacked the C-terminal region to facilitate crystal formation [48]. This high-resolution structure provided valuable details on the protein-DNA interaction, but revealed also numerous discrepancies with the binding modes exhibited by other members of the FOXO family [44], which were attributed to possible crystal-packing issues [48]. For these reasons, the FOXO4-DBD•DBE system offered an excellent opportunity for testing the ability of structural proteomics to probe the conformational effects of binding, which would help rectify or corroborate the observed discrepancies. On the other hand, it also afforded sufficient structural information to determine the validity of the new experimental constraints and evaluate the merits of the selected approaches. 

The experimental strategies were selected for their ability to provide specific information on a typical protein-DNA complex. For instance, HDX was applied to recognize the regions of the protein affected by DNA binding, either through direct protection of the contact interface, or through allosteric conformational changes involving distal regions of the protein. Quantitative XL was used instead to identify possible variations between free and DNA-bound DBD structures, which would help elucidate the effects of the interaction on overall structure topology. In the case of the DNA component, the fast rate of back-exchange characteristic of nucleic acid hydrogens prevented the application of HDX to recognize the surface of the DBE duplex in direct contact with the DBD. As a possible alternative, we explored the application of transplatin (trans-dichlorodiamineplatinum(II), tPt) to generate protein-DNA cross-links that would help locate the mutual positions of interacting structural features [49]. The spatial constraints afforded by these determinations were combined to guide model-building operations and obtain a full-fledged structure for the complex. The results were compared to the available high-resolution structures to assess possible discrepancies and highlight the new information afforded by the selected techniques. It is necessary to point out, here, that the FOXO4-DBD protein construct used in this study was identical to protein constructs used in previous high-resolution structural studies [46,48]. The outcome clearly demonstrated the benefits of structural proteomics to tackle the elucidation of structure and dynamics in systems that elude established high-resolution techniques.

## 2. Materials and Methods 

### 2.1. Materials

Non-labelled and isotope-labelled cross-linkers di(N-succinimidyl) glutarate (DSGd0/DSGd4) and di(N-succinimidyl) suberate (DSSd0/DSSd4) were purchased form ProteoChem (Hurricane, UT, USA). Modified protease trypsin (Gold, mass spectrometry grade) was purchased from Promega (Madison, WI, USA). Nuclease Bal-31 was obtained from New England BioLabs (Ipswich, MA, USA). Liquid chromatography solvents of LC/MS grade were purchased from Thermo Fisher Scientific (Waltham, MA, USA). Other chemicals (highest available purity) were obtained from Sigma-Aldrich. All other chemicals, solvents and buffers for SDS-PAGE were obtained from Bio-Rad Laboratories, Inc. (Hercules, CAUSA). The pET-15b plasmid carrying His-tag, thrombin cleavage site, and FOXO4-DBD (Uniprot ID: P98177-1; residues 86–211) sequences was obtained from Prof. Obsil.

### 2.2. Design of Oligonucleotides and Duplex DNA Preparation

Both, forward and reverse, complementary oligonucleotide strands (5′-TTG GGT AAA CAA G-3′ and 5′-CTT GTT TAC CCA A-3′, respectively) were purchased from Integrated DNA Technologies (USA) in standard desalted purity. The reverse oligonucleotide sequence contained DBE (TTG TTT AC) originating from human NOXA promoter localized on chromosome 18 (+ strand) in the position 59,899,552 to 59,899,564 [50]. Both strands were tested for secondary structure presence to avoid interfering structures. Forward and reverse strands were dissolved in water and mixed in an equimolar ratio and then heated up to 95 °C for 1 min. Next, the mixture was let to cool to the room temperature to form the 13 bp duplex DNA.

### 2.3. FOXO4 Consensus Binding Sequence Determination, Validation and Comparison

The consensus binding sequence of FOXO4 was obtained by searching the HOmo sapiens COmprehensive MOdel COllection (HOCOMOCO) v11 [51], which contains the binding models for 680 human transcription factor (TFs). First, we scanned for TF binding models with a Position Count Matrix (PCM) similar to that of FOXO4 by using HOCOMOCO and Matrix CompaRisOn. Approximate P-value Estimation (MACRO-APE) software [52]. We confirmed that the TGTTT consensus sequence was presented in binding models of other FOX factors. Second, we searched for TF binding models with PCM similar to FOXO4 consensus sequence using HOCOMOCO and MACRO-APE to predict TFs with binding mode highly similar to FOXO4.

### 2.4. Sample Preparation

Full-length DBD (residues 82–207 of the entire FOXO4 sequence) was expressed with an N-terminal His-tag from an appropriate pET-15b plasmid, and then affinity captured on a TALON Superflow Resin (Clontech Laboratories, Mountain View, CA, USA) charged with Co^2+^. The captured protein was submitted to thrombin digestion to eliminate the tag, followed by gel permeation chromatography. A more detailed description of all experimental procedures is included in the Appendix A. The identity, integrity, and purity of the final sample were verified by MS analysis (vide infra). A duplex DNA construct containing one of the DBE consensus sequences (i.e., TTG TTT AC) [42,53] was obtained by annealing complementary oligonucleotides (i.e., 5′-TTG GGT AAA CAA G-3′ forward and 5′-CTT GTT TAC CCA A-3′ reverse). Equimolar amounts were mixed and then heated to 95 °C for 1 min. Finally, the sample was let cool to room temperature to form the 13 bp duplex DNA. The desired FOXO4-DBD•DBE complex was obtained by mixing equimolar amounts of protein and DNA samples in 10 mM HEPES buffer with 50 mM NaCl (pH adjusted to 7.4) to a final 33.7 µM concentration, and incubating the mixture at 18 °C for 1 h. 

### 2.5. Product Characterization

Initial stocks of FOXO4-DBD, DBE construct, and FOXO4-DBD•DBE were analyzed to assess sample purity after expression/purification, annealing of the duplex structure, and proper complex formation. The products of XL reaction were also analyzed in the same fashion to assess the incidence of modification. Briefly, each stock was diluted to a final 10-µM concentration by adding a 7.5 mM solution of ammonium acetate (AA) with 50% MeOH (pH 6.85), and then analyzed on a Bruker Daltonics (Billerica, MA, USA) 15T-Solarix XR Fourier transform ion cyclotron resonance (FTICR) mass spectrometer. FOXO4-DBD, as well as FOXO4-DBD•DBE, were analyzed in positive ion mode. Each sample was loaded onto a syringe and introduced into the electrospray ionization (ESI) source at a 2 µL/min flow rate. The FTICR analyzer was calibrated by using a solution of sodium trifluoroacetate (NaTFA), which afforded a typical 1 ppm accuracy. Mass spectra were acquired over a range of 250–4000 *m*/*z* for 3 min [54]. 

### 2.6. Hydrogen-Deuterium Exchange

HDX reactions were performed on 20 µM solutions of either FOXO4-DBD or FOXO4-DBD•DBE complex prepared in an H_2_O-based buffer (pH 7.4) containing 10 mM HEPES and 50 mM NaCl. After pre-incubation for an hour at 20 °C, the exchange was initiated by diluting each sample 10-fold into a D_2_O-based buffer (pD 7.4) containing 10 mM HEPES and 50 mM NaCl. The reaction was allowed to proceed at 20 °C, while small aliquots containing 100 pmol of protein were taken at predetermined intervals (i.e., 0.33, 2, 5, 10, 30, 60, 180, and 300 min). Quenching was achieved by immediately mixing the aliquot with a 1M glycine/HCl buffer with pH 2.35, and then rapidly freezing the solution in liquid nitrogen. Samples were stored at −80 °C. The analysis was performed by using columns with different immobilized proteases followed by liquid chromatography-mass spectrometry (LC-MS) according to ref. [55]. All experiments were performed in triplicate. A complete description of these procedures is included in the Appendix A (Supplementary Methods) section.

### 2.7. Quantitative Protein-Protein Cross-Linking

Samples containing 20µM of either FOXO4-DBD or FOXO4-DBD•DBE complex in 10 mM HEPES buffer (pH 7.4) with 50 mM NaCl were pre-incubated for an hour at 20 °C before introducing the cross-linking reagent. Separate samples of FOXO4-DBD were treated with either DSGd0 or DSSd0 in their regular, non-labelled form, whereas FOXO4-DBD•DBE samples were reacted with the deuterium-labelled DSGd4 and DSSd4 versions. The reagents were dissolved in dimethyl-sulfoxide (DMSO) to 6.74 mM concentrations and then added to each substrate to achieve a 10:1 molar ratio. The cross-linking reaction was allowed to proceed undisturbed for 2 h, after which corresponding regular and deuterium-labelled samples (e.g., treated with DSSd0 and DSSd4) were mixed in a 1:1 ratio to enable quantification. In parallel, control samples were also examined, which were treated with pure DMSO lacking cross-linker, or matching cross-linker mixtures with a 1:1 ratio of either DSGd0/DSGd4 or DSSd0/DSSd4. All reactions solutions were analyzed by SDS-PAGE to check whether cross-linking had produced any unwanted non-specific higher-order aggregates, or prevented a sufficient degree of digestion necessary to enable subsequent analysis. Characterization of cross-linked conjugates was achieved according to a bottom-up approach that employed trypsin digestion followed by LC-MS analysis [13]. All experiments were performed as triplicate. A complete description is included in the Appendix A.

### 2.8. Protein-DNA Cross-Linking

Samples containing 25µM of either FOXO4-DBD or FOXO4-DBD•DBE complex in 150 mM ammonium acetate (pH 6.85) were treated with a 1 mM solution of trans-platinum(II)diammine dichloride (transplatin, tPt), which had been pre-incubated for an hour at 18 °C. Reaction mixtures containing a final 200 µM concentration of transplatin and 20 µM of protein/complex were incubated at 18 °C for 14 h. In parallel, control samples devoid of transplatin were prepared at the same time in the same manner. Reaction and control samples were analyzed by native and denaturing DNA polyacrylamide gel electrophoresis, SDS polyacrylamide gel electrophoresis, and mass spectrometry. Characterization of cross-linked conjugates was achieved according to a bottom up approach that involved treatment with Bal-31 nuclease and trypsin to digest DNA and protein components, respectively, followed by LC-MS/MS analysis with data-independent acquisition with broad isolation window. A complete description is included in the Appendix A.

### 2.9. Data Processing and Interpretation

The SNAP 2.0 algorithm of the DataAnalysis 4.2 (Bruker Daltonics, Billerica, MA, USA) software package was utilized to generate deconvoluted spectra and lists of monoisotopic masses from the acquired MS data. The MASCOT 2.2 search engine was used to search MS/MS data and achieve the identification of on-line digest products from a theoretical library of digestion products. Deuteration rate was determined by using the home-built Deutex software (unpublished). The home-built LinX software (available online) and Stavrox (v. 3.6.0.1 by Michael Götze) software was then used to compare the experimental data with a library of theoretical cross-linking products to correctly identify the sought-after conjugates. The proportion of labelled versus un-labelled species was determined by applying mMass 5.4.1 [56] to the signals of such conjugates. In the case of peptide-DNA conjugates, deconvoluted spectra and monoisotopic masses were calculated by using a constant unit to mimic the presence of a certain oligonucleotide cross-linked by a transplatin equivalent, which were subsequently searched by LinX. A complete description of these procedures is included in the Appendix A.

### 2.10. Molecular Modeling

Initially, six different FOXO4 sequentially related structures (DBD or DBD•DNA complex) were used as templates for homology modeling of protein part (Modeller software [57]). Such templates, however, covered only the 101–176 residues of the DBD sequence, whereas our selected target covered the 82–207 section including the additional flanking sequences that had previously eluded structural elucidation. The HADDOCK [58] ⁠program was utilized to perform docking experiments between FOXO4-DBD and DBE substrate, where were meant to rationalize the HDX data. The docking experiments utilized a duplex substrate exhibiting an ideal B-DNA conformation, which was generated by the make-na server (http://structure.usc.edu/make-na/). The WeNMR/WestLife infrastructure [59] was used to carry out the computationally intensive docking calculations. The first structure of the cluster of the FOXO4-DBD•DBE complex, which displayed the best HADDOCK score for each run, was used for the subsequent modeling operations. The missing residues in structures of DBD and DBD•DNA were added using the Modeller incorporating distance restraints derived from XL distances. To search the possible conformational space of flanking residues a restrained simulated annealing protocol in the torsion angle space was accomplished in CNS [60]⁠. An ensemble of 50 structures was calculated for each starting model. During simulations, the coordinates afforded by the initial PDB templates and DBE structure were kept fixed, the XL distances (calculated with consideration of the spacer arms lengths [61,62,63] and space requirement of side chains) were used as distance restraints. The resulting models were visualized by using Pymol [64].

### 2.11. Data Availability and Software

MSTools package—available at http://peterslab.org/MSTools [65]LinX—available at http://peterslab.org/MSToolsStavrox (v. 3.6.0.1 by Michael Götze)—available at http://www.stavrox.com/DeutEx—In-house developed program DeutEx is based on a Tcl macro. It requires protein sequence, list of identified peptides from search engines such as MASCOT or PEAKS. Basic overview of the workflow shown on unrelated example data can be found here: http://peterslab.org/downloads/SW/DeutEx.mp4Mass spectrometry data available at https://www.ebi.ac.uk/pride/archive/ (Project accession: PXD013969)

## 3. Results and Discussion

The crystal structure available for the FOXO4-DBD•DBE complex does not cover the entire sequence of the DNA binding domain [48], which spans only the 82–207 section of FOXO4 and omits flanking regions that have been hypothesized to promote the recruiting of additional components of the transcription machinery. At the same time, the NMR structure of full-length FOXO4-DBD provides limited information on the G_138_–A_144_, E_166_–K_170_, and the N- and C-terminal regions, which were described as rather flexible and disordered in solution [46]. Although DNA binding has been credited with stabilizing at least some of these regions, the structure of the bound form still displayed significant discrepancies with those of homologous members of the FOXO family [44], which were possibly caused by crystal packing [48]. For this reason, we investigated such discrepancies by implementing biochemical approaches to probe the effects of ligand binding directly in solution. The study employed recombinant full-length DBD (i.e., residues G_82_–A_207_) and a duplex DNA construct containing the 5′-TAC CCA A-3′ consensus sequence, which was obtained by annealing commercial oligo-deoxyribonucleotides. As shown in Appendix A, mixing equimolar amounts of protein and duplex DNA provided the expected 1:1 species corresponding to the desired FOXO4-DBD•DBE complex. These samples were submitted to hydrogen-deuterium exchange, quantitative protein-protein cross-linking, protein-DNA cross-linking, and docking experiments to obtain complementary information on their mutual interactions and spatial arrangement. The results were compared to those obtained from the individual FOXO4-DBD protein to investigate the effects induced by specific DNA binding.

### 3.1. Combined Online Digestion by Pepsin and Nepenthesin I Improves the HDX Resolution

We initially pursued the identification of the regions of FOXO4-DBD, which were making direct contact with the DBE duplex, or were subjected to detectable microenvironment variations upon binding. Hydrogen-deuterium exchange (HDX) was performed on both free and bound forms of full-length FOXO4-DBD. The determinations followed a well-established protocol in which the exchange process was stopped at predetermined intervals to monitor the rate of exchange. Since hydrogen-deuterium exchange has not yet been widely used to study the complex of protein and duplex DNA, we first tuned conditions for on-line digestion in order to obtain the best spatial resolution. It was achieved using the combination of nepenthesin I and pepsin, where rather small and overlapping peptides were observed. The robustness of the setup was approved by multiple injections. The final analysis was carried out at low pH and temperature to minimize back-exchange. The protocol included protein digestion in consecutive on-line columns containing immobilized proteases (i.e., pepsin and nepenthesin-1), followed by LC-MS/MS and LC-MS analysis of digested peptides (see Materials and Methods and Appendix A) [66]. Since HDX has not been widely used to study complexes comprising protein and duplex DNA, the conditions for on-line digestion required fine-tuning to obtain the best possible spatial resolution. This task was accomplished by combining nepenthesin I with pepsin to obtain rather small, overlapping peptides. Multiple analysis were carried out to evaluate the robustness of this approach. The ensuing peptide map demonstrated that the procedure afforded full coverage of the FOXO4-DBD sequence (Figure 2).

### 3.2. HDX Identified the Interaction Interface and Revealed Long-Distance Structure Stabilization 

For each digestion product, a relative deuteration rate was calculated by considering the number of hydrogens exchanged with deuterium atoms against the total number of exchangeable amide hydrogens in the peptide. Relative deuteration rates versus exchange time were calculated at both the peptide and amino acid levels to recognize possible variations between free and bound FOXO4-DBD (see Appendix A, respectively). This task was facilitated by calculating actual differences for each amino acid in the sequence, which were visualized in a 3D model of FOXO4-DBD•DBE by using an appropriate color palette (Appendix A).

The results are summarized in an HDX difference plot that provided a comprehensive view of the variations of solvent accessibility induced by the specific interactions between FOXO4-DBD and its cognate DBE duplex (Figure 3). Starting from the N-terminus, the G_74_–Y_102_ region displayed relatively high levels of deuteration, regardless of reaction time, with no significant differences between free and bound forms. These observations indicated that this region was rather exposed and capable of exchanging freely with the solvent in both forms. In contrast, the next section spanning the A_103_–T_130_ residues displayed the most extensive differences in deuteration rates, which increased significantly as a function of time. This sequence folds helix H1 and H2, strand S1, and intervening loops (see topology annotation in Figure 2). According to the crystal structure, none of these distinctive features is supposed to make direct contact with the duplex DNA [48], which would help explain the drop in deuteration by invoking a simple protection effect. In the absence of direct contact, the observed loss of solvent accessibility must be attributed to indirect conformational effects induced by binding. The fact that the difference in deuteration levels increased gradually with time and stabilized after 30 min suggests that, in the free form, this set of secondary structures may undergo slow mutual dynamics that delay the exchange of susceptible hydrogens. In the bound form, such dynamics may be stabilized by interactions with contiguous structures that, in turn, make direct contact with the DNA ligand. Like falling dominoes, a series of relatively minor conformational variations linked together may ultimately induce observable inhibition of the exchange reaction. This long-distance effect is clearly evident, for example, in the relative deuteration plot of peptide A_103_–L_118_ (2–3), which shows increasing uptake in the free FOXO4-DBD as a function of time, but constant low-level deuteration in the bound form (Appendix A).

The next region, spanning the V_131_–K_159_ residues, also manifested significant differences between free and bound forms, but their time dependence displayed a rapid increase at shorter intervals, followed by a decline near initial levels at longer reaction times (Figure 3A). In particular, the residues forming helix H3, which the crystal structure places directly in the major groove of the DBE duplex [48], experienced the largest differences. For this reason, direct steric protection induced by bound DNA could explain the uptake inhibition observed for such residues. In contrast, the outcome observed for the contiguous helix H4 and intervening loop could indirectly result from the stabilization of the H3 conformation, which could constrain the placement of such residues and restrict their solvent accessibility. The crystal structure also identified a handful of contacts that were mediated by water molecules trapped in the binding interface. While these interactions have been suggested to further stabilize the dynamics of helix H3 and flanking regions, it is not clear how trapping D_2_O versus H_2_O present in the solvent may affect the observed deuteration rates. It should be also pointed out that FOXO4 differs from other FOXO homologues by the insertion of five amino acids (K_137_–N_141_) in the H4–H3 loop (Figure 2). It is not clear whether this insertion may be responsible for the unusual conformation assumed by helix H3 in the bound form, which differs from those assumed in other members of the family [48]. In any case, the observed inhibition pattern supports a role in stabilizing the fold of FOXO4-DBD and reducing its overall flexibility upon complex formation.

The F_160_–D_195_ section corresponds to sequences located on the C-terminal side of helix H3, which manifested smaller but still perceptible differences of deuteration rates. This region contains strands S2 and S3, as well as the W1 and W2 wings, which the crystal structure placed far removed from the DNA binding interface. Also in this case, an overall decrease of structural flexibility upon binding could explain the reduced deuterium uptake. It should be noted that, in addition to conferring FOXO4-DBD its winged look and fine tuning of the interaction with DNA [47], W1 and W2 could contribute to constitute possible regions of contact for auxiliary components of the transcription complex. The final section covered by HDX determinations consisted of the S_196_–A_207_ sequence and displayed marginal deuteration differences. This observation indicated the absence of any protection or conformational effects induced by DNA binding.

### 3.3. Protein-DNA Cross-Linking Revealed the Mutual Placement of Protein and DNA Components

The HDX experiments identified the regions affected directly and indirectly by DNA binding, which experienced clearly detectable variations of solvent accessibility. However, these types of determinations could not identify the structures responsible for limiting the access of solvent to a specific region. In other words, these experiments could not reveal the mutual spatial relationships between such structures, nor assess the effects of binding on such relationships. For this reason, we employed different types of cross-linking strategies to probe the organization of the various structures and recognize their mutual placement in the overall fold. The first approach employed transplatin to generate putative protein-DNA conjugates that may be capable of constraining the position of the DBE ligand onto the FOXO4-DBD substrate. The reactivity of platinum compounds towards specific functional groups of nucleic acids is well documented [67] and involves the preferential attack of the N7 position of guanine base [68]. Although the characteristics of their reactivity towards protein residues are still unclear [49], amino acids with electron-rich S, N and O atoms, such as Cys, Met, His, and Thr, have been described as preferred targets [69]. We treated samples of FOXO4-DBD•DBE complex, as well as free FOXO4-DBD and DBD, with a 10:1 transplatin to substrate molar ratio in 150 mM ammonium acetate (pH 6.85) and incubated at 18 °C for 14 h (see Materials and Methods and Appendix A). The sample mixtures were analyzed by both ESI-MS and SDS-PAGE to assess the distribution of transplatin adducts to estimate the proportion of sought-after intermolecular cross-links. The representative data in Figure 4A provides a view of the typical product distributions obtained from these probing reactions, which included monofunctional “dangling” adducts containing a still unreacted chloride function (i.e., marked as tPt-Cl adducts), as well as bifunctional conjugates in which both functions had effectively reacted (i.e., marked as tPt adducts).

Based on mass alone, it is not typically possible to distinguish the desired intermolecular conjugates from intramolecular crosslinks, which share the same elemental composition. For this reason, ESI-MS analysis was repeated in the presence of 50% methanol to achieve mild denaturing conditions. In this way, products stabilized by bridging bifunctional cross-links were still detected intact, such as the conjugates containing FOXO4-DBD and individual DBE-F or DBE-R strands. In contrast, no signal was observed for complexes devoid of any intermolecular conjugation, whereas adducts of their free unbound components were individually detected, such as those of FOXO4-DBD protein, DBE-F, and DBE-R strand (Figure 4B). In analogous fashion, the reaction mixtures were also analyzed by PAGE under both native and denaturing conditions (Appendix A). Direct data comparison enabled the identification of bands that eluded dissociation by elevated concentrations of SDS or urea and, thus, could be attributed to the presence of bridging bifunctional crosslinks. These data enabled us to estimate that the desired intermolecular cross-links amounted to less than 5% of the total material submitted to transplatin reaction.

A classic bottom-up strategy was carried out to complete the characterization of cross-linked products, which included digesting the material with protein- and nucleic acid-specific enzymes to obtain samples amenable to LC-MS and LC-MS/MS analysis. In particular, reaction mixtures were treated with trypsin to map the position of peptides conjugated to DNA strands (see Materials and Methods and Appendix A). In subsequent experiments, the size of the oligonucleotide moieties was reduced by treatment with Bal-31 nuclease to facilitate analysis. The representative data in Appendix A of Appendix A illustrates the challenges faced by the MS/MS analysis of these types of hetero-conjugates. Upon gas-phase activation, a precursor ion consisting of G_74_–R_88_ cross-linked to the DBE-R strand underwent dissociation around the bridging Pt atom, rather than along the backbones of the bridged moieties. The absence of sequence information afforded by this type of fragmentation prevented the identification of the actual residues involved in the cross-linking reaction. Nevertheless, the identity of the conjugated components still represented valuable information on the mutual relationships between contiguous regions (summarized in Appendix A).

The detected peptide-oligonucleotide conjugates were examined in the context of the results afforded by the HDX determinations and other structural information available for the system. For instance, peptide N_148_–K_159_ spanning helix H3 was found conjugated to the forward strand of DBE, consistent with the placement of H3 directly into the major groove of the DNA duplex in the crystal structure [48]. This finding agreed also with the prominent protection effects observed in this region during HDX experiments (Figure 3A). Peptide F_160_–K_170_, covering the end of S2 and beginning of W1, was also cross-linked to the forward strand of DBE, despite the absence of any direct contact in the crystal structure. In this peptide, reactivity and orientation considerations would point towards H164 and T168 as possible conjugation sites, if their distances from susceptible DNA structures were sufficiently favorable. In this direction, the HDX data indicated that this region experienced a detectable decline in deuterium uptake consistent with the adoption of a rather constrained conformation upon DNA binding (Figure 3A). The new conformation could place susceptible groups within mutual striking distance, thus promoting the formation of the observed cross-linked product. A similar explanation is applicable also to the S_171_–K_182_ peptide spanning the end of W1, beginning of W2, and intervening S3 strand, which formed cross-links with both forward and reverse strands of DBE. Also, this region experienced a significant decrease of deuterium exchange upon binding, which was not explainable by direct steric protection, but rather by indirect conformational effects transmitted through contiguous structures. The crystal structure orients S_171_ and S_172_ to face the minor groove of the duplex construct, which would represent prime positions for promoting conjugation with either strand. Also in this case, the respective functional groups could be placed within striking distance by the more constrained conformation revealed by HDX experiments. The remaining products consisted of the G_74_–R_88_ peptide cross-linked to either the forward or reverse strand. These products address the flexibility of the N-terminal loop, which was supported by the lack of any significant variation of deuteration patterns reported by the HDX experiments.

### 3.4. DNA Binding Induced Significant Effects on Protein Conformation

The observed protein-DNA cross-links provided valuable information not only on the reciprocal positions of protein and DNA components, but also on the significant changes induced by binding on the initial protein conformation. We employed protein-specific reagents to evaluate the extent of such variations and enable a better appreciation of indirect conformational effects. Our quantitative crosslinking approach involved the concerted application of the homobifunctional reagents DSGd0/d4 and DSSd0/d4 [di(N-succinimidyl) suberate and di(N-succinimidyl) glutarate] to bridge susceptible amino or hydroxy groups that may be respectively placed within 20.5 ± 3.0 or 24.2 ± 3.0 Å of one another (see Appendix A). The utilization of reagents with different bridging spans provided the ability to determine an average distance between residues. At the same time, the isotopic labels facilitated the identification of cross-linked products in complex digestion mixtures from their characteristic 4-Da spacing and enabled the acquisition of unbiased quantitative data on the incidence of cross-linking in the free or bound FOXO4-DBD. Initially, separate samples were treated with 1:1 mixtures of matching unlabeled/labelled reagents of the same length to complete a survey of the regions susceptible to cross-linking (see section Materials and Methods and Appendix A). A total of 39 conjugates were identified, which bridged lysine and serine residues, as well as the N-terminal amino group (Appendix A). The majority of them were detected in matching pairs generated by reagents of either length, and were observed in both free and bound samples. However, a small portion was unique to just one form and/or cross-linker length. Next, individual aliquots of free FOXO4-DBD were treated with either DSGd0 or DSSd0, whereas those of FOXO4-DBD•DBE complex were separately treated with either DSGd4 or DSSd4 (see Materials and Methods and Appendix A). Corresponding samples treated with the same unlabeled/labelled reagent were mixed in a 1:1 molar ratio prior to protease digestion and analysis to compare the incidence of each conjugate in either free or bound samples. The proportion of each of the 39 conjugates identified earlier was determined for at least one of the cross-linker lengths, as summarized in (Appendix A).

A close examination of the results revealed distinctive cross-linking patterns associated with the presence of bound DNA. In particular, the incidence of some conjugates decreased significantly upon binding, while others increased. Among the former, the DGS conjugates bridging K_147_ to either K_170_ or N-term dropped from 95.9% and 92.8% to 4.1% and 7.2%, respectively (Appendix A). The cross-linking inhibition manifested by K_147_ cannot be merely ascribed to its location on helix H3, in direct contact with the duplex construct, because this residue was still capable of supporting conjugation with both K_135_ and K_137_. A more plausible explanation could be that DNA binding forced K_147_ out of the reach of either K_170_ on helix H2, or K_162_ on the adjacent S2 region. The limited nature of such conformational changes was revealed by the fact that K_147_ was pushed out of N-term’s reach for the shorter DSG reagent, but was still sufficiently close for the longer DSS, with incidence of cross-linking dropping from 92.8 to 7.2% and from 72.7 to 27.3%, respectively (see Appendix A). The limited extent of these changes was also evident in the subtler cross-linking variations between K_147_ and either K_135_ or K_137_ located on the H3-H4 intervening loop.

In other cases, DNA binding increased the incidence of specific conjugates by placing residues within mutual striking distance in the complex, which were marginally susceptible or inert in the free protein. For example, the conjugates bridging residue K_182_ with N-term, K_89_, K_116_, K_159_, K_162_, or K_182_ were greatly enhanced by the presence of DNA duplex (Appendix A). For the majority of these positions, the levels of cross-linking observed with the longer DSS reagent displayed more significant variations than those with the shorter DSG. Considering that K_182_ is located on the W2 wing region, these observations offered further evidence of the long-range conformational effects of DNA binding suggested by the results of HDX and protein-DNA cross-linking experiments. Another example was provided by the numerous conjugates involving the N-term, which suggested that DNA binding had prominent stabilizing effects on a region that was rather flexible in free FOXO4-DBD. Consistent with the HDX data, the variations of cross-linking patterns confirmed that DNA binding induced significant effects on protein conformation not only within the contact interface, but also in rather distal positions.

### 3.5. Structural Proteomics Could Effectively Guide Model-Building Operations to Produce Very High-Quality 3D Models

The HDX and XL experiments provided a wealth of information that was used to guide the molecular modelling of a full-fledged FOXO4-DBD and FOXO4-DBD•DBE complex. Our approach took advantage of available high-resolution structures that, although incomplete in their coverage of the protein sequence, still represented excellent templates for homology modelling operations (see Materials and Methods and Appendix A). In particular, the templates were used to obtain the coordinates of what could be defined as the structured core of the complex, a region spanning approximately from R_93_ to N_177_, which displayed limited discrepancies across the available structures. In contrast, the regions that were either absent from the templates, or displayed significant variations, or had been predicted to possess a high degree of flexibility by PSIPRED [70], were modeled according to the HDX and cross-linking information (see Materials and Methods and Appendix A). These regions corresponded to the G_74_–Q_100_ and N_177_–A_207_ sections located respectively at the N- and C-terminal ends of the DBD sequence. These operations were performed in the Modeller suite [57], which was also used to eliminate possible strains and steric clashes introduced during model building. The program applied the DOPE scoring algorithm to identify the best possible structures that were subsequently employed in docking and simulated annealing procedures. The former was carried out to place the DBE structure, which was created separately by using the make-na server (http://structure.usc.edu/make-na/), onto the putative binding site of the protein. This operation was accomplished in HADDOCK [58] by designating as active those residues that had experienced reduced rates of exchange upon DNA binding (Figure 3A and Appendix A). The mutual positioning between the DBE and FOXO4-DBD components was further refined according to the results of the protein-DNA cross-linking experiments, which were introduced by using Modeller. Finally, the structures of both FOXO4-DBD and FOXO4-DBD•DBE complex were submitted to simulated annealing and energy minimization in CNS [60] to generate the sought-after model ensembles (Figure 5).

The structures obtained for FOXO4-DBD and FOXO4-DBD•DBE complex were compared to the corresponding high-resolution structures to assess the robustness of our structural proteomics approach. In the case of individual FOXO4-DBD, the overall topology of the ensemble reflected the typical forkhead structure of the FOXO family and matched very closely that of the NMR structure used as homology template (see Appendix A), thus supporting the validity of the HDX and cross-linking constraints. A more detailed comparison was obtained by calculating the root mean square deviation (RMSD) between the coordinates of corresponding heavy atoms located in the backbone of each ensemble model and the various templates. The representative plot in Appendix A, for example, shows that the model obtained from the 1e17 structure deviated very little from the initial template. The fact that the experimental constraints introduced during modelling did not force any significant variation of the initial coordinates indicates that the probing operations did not cause any perturbation of the substrate’s 3D structure and corroborated the excellent stability of the structured core of FOXO4-DBD. In contrast, larger RMSD values were obtained when models based on other templates were compared with the initial FOXO4 models used in the study (3L2C, 1E17), as expected from the discrepancies between the various NMR and crystal structures available (see Appendix A). Although this type of analysis was not possible for the regions that were absent from the templates, the excellent match manifested by the regions present in both model and template warranted a high level of confidence in the entire structures produced by our approach.

The ensemble of the FOXO4-DBD•DBE complex was examined in similar fashion. Also in this case, the overall topology matched very closely that of the corresponding high-resolution template (i.e., 3l2c), with the DBE component oriented in the proper direction and placed in the correct position onto the FOXO4-DBD’s binding site (see Appendix A). RMSD comparisons between the model and crystal structure revealed excellent match for the regions present in both, thus ruling out the possibility of inadvertent perturbations introduced by the probing procedures. Additionally, we determined the distances between residues that had been conjugated by the cross-linking reagents, and then compared them with the corresponding distances measured on the crystal structure. The resulting RMSD values revealed excellent agreement across the board, with the sole exception of the distances between the DBE molecule (DBE-F) and specific residues of the H_164_–M_175_ loop (H_164_, T_168_, S_171_, S_172_ or M_175_), which were somewhat longer in our model (see Appendix A). These discrepancies; however, were consistent with the high degree of flexibility possessed by the loop, which was manifest also in the higher B-factors displayed by this region in the crystal structure. In agreement with the crystal structure, the models showed that helix H3 represents the main interaction interface, as indicated by both HDX and cross-linking data (see Appendix A). The DBE structure employed here replicated the consensus binding sequences for all related FOX factors, which contain a general TGTTT motif surrounded by more variable sequences (see Appendix A). Whereas FOXO3 and FOXO6 recognize two nucleotides located after this consensus sequence, FOXP1 only recognize the second nucleotide but not the first one. In contrast, our model indicates that FOXO4 may recognize one nucleotide before and one after the consensus motif, thus affording additional evidence of the uniqueness of the interactions established by this member of the FOXO family. These results were supported also by Position Count Matrix (PCM) values that estimated the binding probability of individual bases at each position in the sequence (see Appendix A).

A close comparison of the structures of FOXO4-DBD and FOXO4-DBD•DBE complex obtained by our approach allowed us to further explore the effects of binding on protein conformation. The examination confirmed that the N- and C-terminal sequences remained largely unstructured even after DBE binding, as represented by the mesh regions of our models (Figure 5). The main interface region consisting of H3 showed limited variations between unbound and bound forms. Similar outcomes were also observed for the contiguous H1–H2 loop and H2 helix. In contrast, loop H2–H4–H3 and the S2 and S3 strands of wing W1 showed rather large variations upon binding. Additionally, also the N- and C- terminal regions manifested extensive variations. These observations were consistent with the HDX data that revealed clearly peculiar time dependencies. For instance, the initially increasing rates in the H1 helix, H1–H2 loop, and H2 helix (A_103_–T_130_, in particular) suggested variations of dynamics upon DBE binding, whereas their subsequent decreasing rates were consistent with the actual structural stabilization resulting from the presence of bound DNA. The RMSD values calculated for corresponding heavy backbone atoms provided an excellent measure of these conformational effects (see Appendix A). The values obtained from flanking regions near the interface, indicating major changes in loop H4–H3 and W1 wing and minor changes in loops H1–H2 and H3–S1, highlighted the indirect effects of binding, which were consistent with the results of HDX and XL experiments. The conformational changes revealed by this type of treatment were consistent with a classic adaptive binding mechanism by which rather sizeable conformational changes may be necessary to establish specific substrate-ligand interactions. The fact that the observed conformational changes were not limited only to the sequences in direct contact with the ligand DBE, but involved also contiguous regions, supports mechanisms by which binding events may trigger associated activities through allosteric effects, or place bordering regions in positions necessary to mediate the recruiting of additional factors, such as the acetyl transferases that are known to interact with other members of the FOXO family [50,71].

## 4. Conclusions

The structural investigation of FOXO4-DBD and FOXO4-DBD•DBE provided a thorough assessment of the ability of structural proteomics techniques to obtain valid information on systems that are not directly amenable to classic high-resolution approaches. The outcome showed that the concerted application of HDX and XL could effectively guide model-building operations to produce high-quality 3D models. Our multi-step strategy involved the utilization of high-resolution templates to carry out initial homology modelling. The sections that were not present in the templates were generated from experimental constraints and integrated with the initial structures to cover the entire DNA-binding domain. Although the high-resolution templates did not cover the entire structure folded by our construct, they still provided sufficient overlap to enable an unbiased assessment of the validity of the results afforded by our experimental/computational workflow. In fact, the excellent match between the templates and our structures ruled out the possibility that cross-linking procedures might have introduced unwanted artefacts or structure distortion. Further, an agreement between the templates and the corresponding portions of our models ruled out such possibilities and confirmed the ability of the selected computational strategies to translate these types of experimental constraints into actual 3D structures. Validating the approach on the “known” portions of the structures was essential in supporting the validity of the “unknown” sections that were conspicuously absent from the templates. The fact that the results of HDX and cross-linking experiments were in consistent mutual agreement provided further proof of the robustness of our concerted approach. For these reasons, our models represent comprehensive structures of full-fledged FOXO4-DBD and FOXO4-DBD•DBE.

The pictures painted by the high-resolution templates (identical protein construct was used), which were obtained by NMR and crystallography, are not only incomplete, but in the case of crystallography also static. Our extensive data provide a wealth of new information on the conformational dynamics of the protein in both unbound and bound forms. Our experiments clearly differentiated regions that were conformationally stable from those that underwent significant conformational changes upon DNA binding. The most important finding was that binding affected not only the interface region, but also the conformation of regions that were located away from the interface. This information might be essential to understand the allosteric properties of the complex and their role in recruiting additional transcriptional factors.

Our models confirmed that full-length FOXO4-DBD adopts the classic forkhead topology characteristic of this family of transcription factors, but corroborated also its unusual DNA binding mode that is unique among those manifested by the highly homologous FOXO proteins [44]. The close match between our model and the crystal structure of FOXO4-DBD•DBE ruled out the possibility that crystal packing might be the cause of the significant differences noted between the types of interactions established by FOXO4 versus those involving the other members of the family [48]. Our models confirmed the prominent role of helix H3 in such interactions and highlighted the involvement of neighboring regions, which may be responsible for fine-tuning the sequence-specific recognition of the correct DNA counterpart.

In conclusion, this study demonstrated the merits of structural proteomics approaches for the elucidation of protein-nucleic acid complexes. The utilization of specific hydrolytic procedures to complete the characterization of HDX and cross-linking products virtually eliminates any limitation pertaining the size of the species of interest. Propelled by continued advances in the computational approaches employed to translate the experimental results into all-atoms models, structural proteomics has rapidly emerged as a valid complement, and often alternative, to the classic high-resolution techniques. By clearly demonstrating the applicability to transcription factor-response element complexes, we hope that this study will lead to a broader utilization of structural proteomics to mitigate the chronic dearth of information on these essential regulatory systems.

## Figures and Tables

**Figure 1 biomolecules-09-00535-f001:**
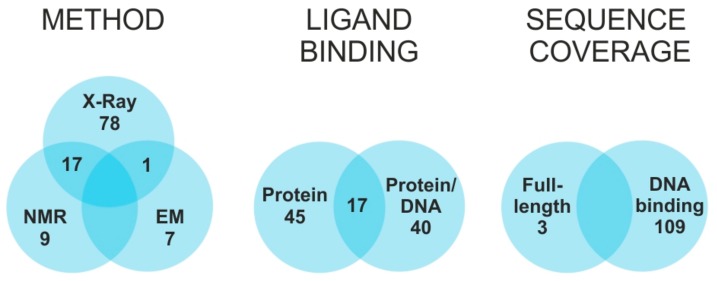
Transcription factors structures—current state: Statistics on high-resolution structures deposited in the Protein Data Bank (80), with consideration of selected methodology, presence of interaction partner, and sequence coverage. To date, the structures of only 112 human TFs have been solved out of a total predicted to be in the 1300–1900 range [6].

**Figure 2 biomolecules-09-00535-f002:**
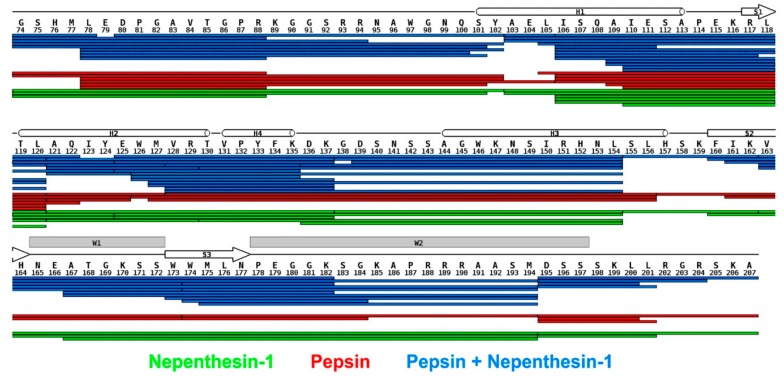
Combined online digestion by pepsin and nepenthesin I improves the HDX resolution. Comparison of peptide maps obtained by on-line proteolysis with nepenthesin-1 (green), pepsin (red), or their combination (blue) is projected on the FOXO4 sequence. In all cases, full sequence coverage was reached but pepsin/nepenthesin-1 digestion provided the highest redundancy and spatial resolution. Secondary structure elements are depicted above the sequence. The N-terminal region G_74_–P_87_ originates from the production plasmid and thus is not a part of the wild-type FOXO4 sequence (see also Appendix A).

**Figure 3 biomolecules-09-00535-f003:**
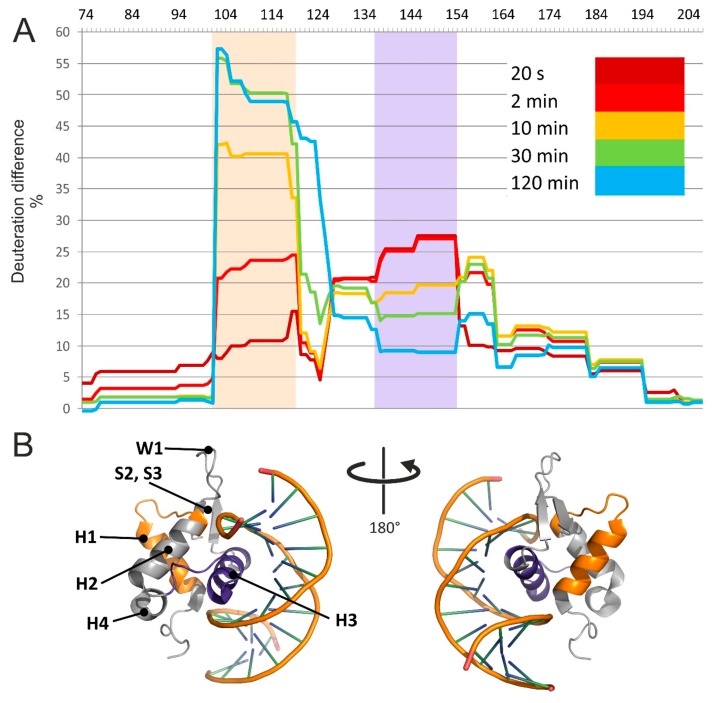
HDX identified the interaction interface and long-distance stabilization of the protein structure. (**A**) Relative deuteration differences [DR(FOXO4-DBD) – DR(FOXO4-DBD•DBE)] plotted along the FOXO4-DBD sequence and their evolution in time. Highlighted areas show two regions with large differences in deuteration levels. (**B**) FOXO4-DBD•DBE structure with highlighted regions showing significant differences in deuteration. Please note that the G_74_–P_87_ region of the construct was contributed by the recombinant-production vector and, thus, was not part of the wild-type FOXO4 sequence (see also Appendix A).

**Figure 4 biomolecules-09-00535-f004:**
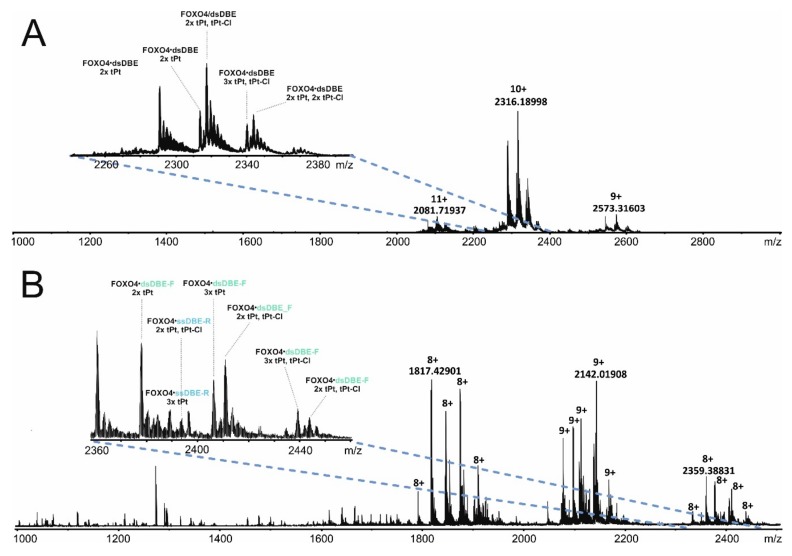
Mass spectra of transplatin-treated FOXO4-DBD•DBE samples, which were obtained under either (**A**) non-denaturing (i.e., 1% acetic acid in 7.5 mM ammonium acetate) or (**B**) denaturing (i.e., 1% acetic acid and 50% methanol in 7.5 mM ammonium acetate solution) conditions. Abbreviations: tPt—bifunctional transplatin conjugate (Pt(NH2)_2_); tPt-Cl—monofunctional adduct (Pt(NH2)_2_Cl); dsDBD—duplex DNA; ssDBD-F—forward oligonucleotide strand; ssDBD-R—reverse oligonucleotide strand. The FOXO4-DBD sample employed in these experiments lacked three amino acids at the C-terminus (see also Appendix A.

**Figure 5 biomolecules-09-00535-f005:**
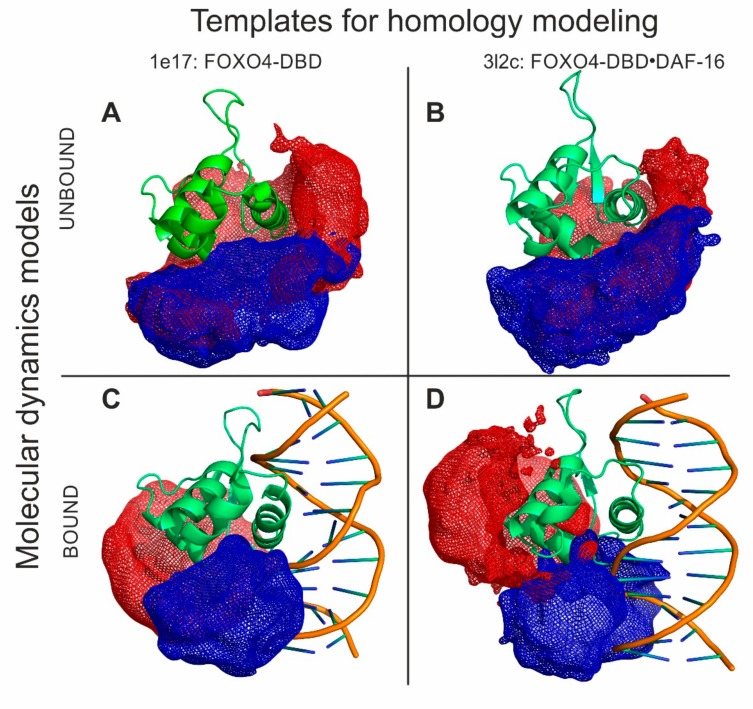
Structural proteomics could effectively guide model-building operations to produce very high-quality 3D models. Models of FOXO4-DBD and FOXO4-DBD•DBE were obtained by combining homology modelling with experimental constraints and molecular dynamics simulations. These models incorporated extensive information from protein-DNA cross-links, quantitative protein-protein cross-links, and hydrogen-deuterium exchange. The green structures show representative models for unbound (**A**,**B**) and bound (**C**,**D**) forms based on corresponding 1E17 (**A**,**C**) or 3L2C (**B**,**D**) high-resolution templates. Mesh areas in blue and red colors represent spaces occupied by all the models in the ensembles, which provided a measure of the flexibility of the N- and C- terminal regions (see also Appendix A).

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
