# Peer review of "MS-Based Approaches Enable the Structural Characterization of Transcription Factor/DNA Response Element Complex"

_biomolecules, 2019, doi:10.3390/biom9100535_

Round 1

Reviewer 1 Report

The manuscript entitled “MS-based approaches enable the structural characterization of transcription factor / DNA response element complex” by Slavata et al, employed mass spectrometry/HDX based information to reconstruct the FOXO4-DNA model that is incomplete and inconsistent with other FOXO family members. In general, the manuscript highlights the potential of proteomic techniques to complement the computational methods and offer new insights. The method is strong, and the manuscript is well written.

I have few suggestions for the authors.

Comments

As per the figure 2, the modelled structure is entirely loop region that explain why crystallography could not capture it. Did the author calculate the secondary structure propensity of this region? Why the DNA has been constructed from online server rather than taking it from the crystal structure. The crystal structure should represent the more natural structure rather than creating a perfect B-type DNA. How did you select the DNA flanking region? The pdb ID 3L2C had different DNA sequence than the authors used in this study. Line 313: “Note that the G74 - P87 region of the construct was contributed by the recombinant-production vector” then the DBD should start from 88? Line 657: “NMR and crystallography are not only incomplete, but also static.” Is not very accurate. NMR is usually give an ensemble of models. The label for figure 5 is Figure 3; please correct it in the revision. How the representative protein model in Figure 5 has been selected? In abstract, at line 28, “arematerial” may be a spelling mistake. Other than this, e.g, nepentesin-1 at line 311 is not correct. Figure S28 (C) has been generated after docking, I guess, but it has not been mentioned. There is a visible difference between S9 (protein model part) and S28 (part C). How did Fig S9 has been generated? In HDX figures (fig S3-5), what is “F” and “FD” in mentioned in the figures, as well as please explain the parenthesis figures written along the amino acid length. Figure S7, mapping of deutration rate on proteins; it seems that the protein structures are from crystal not from the modelled one. If this is the case, how the results are related to complete FOXO-DBD? Also, it is hard to see the color difference on a black background.

Reviewer 2 Report

In this manuscript Slavata et al. report the application of HDX and cross-linking MS for analysis of a protein-DNA complex. Utilization of HDX and cross-linking MS is clearly an important strategy to obtain structural information for biomolecular complexes. However, I’m unclear about the focus of the manuscript in its current form, in particular if it should be method-oriented or biological-oriented.

Given the presented data, the authors by far exaggerate their main conclusions. They clearly oversell the role of MS in analysis of structure and dynamics of biomolecules. For example, they claim that NMR provides only static pictures for the FOXO4-DBD which is clearly not the case. NMR spectroscopy is the only experimental technique that can provide insight into the timescales and amplitudes of motions over a wide range of time scales from ps/ns to days with atomic resolution. MS can in the best case provide a semi-quantitative insight into ‘dynamics’ (only protection of exchangeable protons!). The advantage of MS compared to NMR and EM/X-ray is that it can provide valuable insight into biomolecular interfaces and ‘dynamics’ using smaller sample amounts and for large and challenging biomolecular complexes. The FOXO4-DBD is clearly not such a system If the authors aim to focus on the validity of MS to obtain insight into ‘dynamics’ (methods paper) they should compare their results to the current gold standard NMR (H/D exchange, 15N spin relaxation etc.). In its current form the manuscript reads quite lengthy, including a lot of interpretations/speculations which are invalid based on the available data. I’m wondering how reproducible the MS HDX/cross-linking data is. Can the authors provide an error estimate/error bars or any meaningful statistics/validation on the data shown?

Minor comments:

Line 603: ‘FOXP1’ should be ‘FOXO1’

Author Response

Rev 2.

In this manuscript Slavata et al. report the application of HDX and cross-linking MS for analysis of a protein-DNA complex. Utilization of HDX and cross-linking MS is clearly an important strategy to obtain structural information for biomolecular complexes. However, I’m unclear about the focus of the manuscript in its current form, in particular if it should be method-oriented or biological-oriented.

Given the presented data, the authors by far exaggerate their main conclusions. They clearly oversell the role of MS in analysis of structure and dynamics of biomolecules.

            The structural proteomics approach (chemical cross-linking, hydrogen/deuterium exchange and native electrospray) have become very popular analytical techniques for protein structural modeling and multi-protein assembly characterization. Even the concept is being almost two decades adopted by several well-established research groups; the potential of such technology has never been tested to interrogate structural properties of protein-dsDNA complexes. Thus our group initiated experiments where the FOXO4/DAF16 complex was characterized by hydrogen/deuterium exchange, quantitative chemical cross-linking and protein-dsDNA cross-linking. The DNA binding domain of FOXO4 transcriptional factor was recombinatly expressed in E. coli and its homogeneity was confirmed by mass spectrometry and gel-filtration. Native electrospray and gel shift assay proved its ability to bind DAF16 motif (double stranded DNA representing FOXO4 DNA response element). Initial hydrogen/deuterium experiments on apo and holo forms (in the presence of DAF16) of FOXO4 successfully identified regions on FOXO4 responsible for DAF16 binding. Subsequent quantitative chemical cross-linking experiments and protein-dsDNA experiments shed a light on the dynamic structural rearrangement upon DAF16 binding. All experimental data were finally utilized for in silico homology modeling and molecular docking which enable us to create structural models for apo and holo forms of FOXO4 DNA binding domain. Finally, both structural models were compared with high resolution structural models (NMR structural model for apo and X-ray structural model for holo) and very nice overlaps were achieved. Moreover, our experimental workflow allowed structural characterization of intrinsically disordered parts of FOXO4 binding domains as well in contrast to conventional high resolution techniques. It is worth to stress out that such study is the first example where mass spectrometric techniques are capable to characterize transcription factor/DNA response element complex

For example, they claim that NMR provides only static pictures for the FOXO4-DBD which is clearly not the case. NMR spectroscopy is the only experimental technique that can provide insight into the timescales and amplitudes of motions over a wide range of time scales from ps/ns to days with atomic resolution. MS can in the best case provide a semi-quantitative insight into ‘dynamics’ (only protection of exchangeable protons!). The advantage of MS compared to NMR and EM/X-ray is that it can provide valuable insight into biomolecular interfaces and ‘dynamics’ using smaller sample amounts and for large and challenging biomolecular complexes. The FOXO4-DBD is clearly not such a system.

We agree, that this generous statement about static pictures provided by NMR is not correct and it was change according reviewer concerns.

If the authors aim to focus on the validity of MS to obtain insight into ‘dynamics’ (methods paper) they should compare their results to the current gold standard NMR (H/D exchange, 15N spin relaxation etc.).

You are totally right, but it would take lot of time, approximately 1-2 years.

In its current form the manuscript reads quite lengthy, including a lot of interpretations/speculations which are invalid based on the available data.

I’m wondering how reproducible the MS HDX/cross-linking data is. Can the authors provide an error estimate/error bars or any meaningful statistics/validation on the data shown?

Yes. Both, HDX and crosslinking experiments, were done in triplicates. For crosslinking, the average values with standard deviations are summarized in Tab. S2. For HDX, the triplicates are in Figs S3 to S5 represented by medium value and error bars.

Minor comments:

Line 603: ‘FOXP1’ should be ‘FOXO1’

This is not a mistype. We really used FOXP1-DBD which shows a 51% sequence identity with FOXO4-DBD.

Reviewer 3 Report

This manuscript describes a set of experiments including hydrogen-deuterium exchange, protein-DNA crosslinking, and subsequent analysis with mass spectroscopy and molecular modeling of the protein-DNA complex. This manuscript will be of great interest to those studying or interested in protein-DNA binding and concomitant protein confromational changes with binding. The manuscript was well written with an understandable approach and copious comparison to literature. My few critiques are numbered below.

Line 28: "...arematerial..." I assume this is a misprint or error of some kind. I suggest removing it and replacing it with something else that conveys the authors' intent. I think I understand what the overlap regions in Figure 1 represent, but the authors should be more clear in the Figure 1 figure caption. Did the authors ever measure the binding constant of FOXO4-DBD with DBE, such as with ITC to not only gain perspective on the strength of the interaction, but on the enthalpic and entropic components of the interaction. How do these values support your proposed conformation for the complex? Figure 3. In part B of the figure caption, it states highlighted regions are incorporated in the figure showing significant differences in deuteration levels. I cannot see any highlighted regions. Could they correspond to the highlighted regions in part A? If so, those colors from A are not visible in the images of part B.

Author Response

Rev 3.

This manuscript describes a set of experiments including hydrogen-deuterium exchange, protein-DNA crosslinking, and subsequent analysis with mass spectroscopy and molecular modeling of the protein-DNA complex. This manuscript will be of great interest to those studying or interested in protein-DNA binding and concomitant protein conformational changes with binding. The manuscript was well written with an understandable approach and copious comparison to literature. My few critiques are numbered below.

Line 28: "...are material..." I assume this is a misprint or error of some kind. I suggest removing it and replacing it with something else that conveys the authors' intent.

We agree with this point. Appropriate changes have been made.

I think I understand what the overlap regions in Figure 1 represent, but the authors should be more clear in the Figure 1 figure caption.

We agree with this point. Appropriate changes have been made.

Did the authors ever measure the binding constant of FOXO4-DBD with DBE, such as with ITC to not only gain perspective on the strength of the interaction, but on the enthalpic and entropic components of the interaction. How do these values support your proposed conformation for the complex?

Since the binding constant of FOXO4-DBD with DBE (Kd=93.4+/- 7 nmol/L) was determined using fluorescent anisotropy in the previous by Boura et al (Fig. 1C, DOI: 10.1158/0008-5472.CAN-10-2203, we did not measure it.

Figure 3. In part B of the figure caption, it states highlighted regions are incorporated in the figure showing significant differences in deuteration levels. I cannot see any highlighted regions. Could they correspond to the highlighted regions in part A? If so, those colors from A are not visible in the images of part B.

Colors in part A correspond to colors in part B. Orange rectangle in A is represented by orange helix H1 and subsequent loop in B. Purple rectangle in A is represented by helix H3 in B.

Reviewer 4 Report

The authors presented MS-based approaches for the structural characterization of dsDNA-protein complexes; in this case, the transcription factor FOXO4 and DBE. In general, this article seems to be geared toward creating a new method to evaluate the structural insights towards the binding mechanism of flexible proteins. It is obvious from the manuscript that most of the data generated was used to support existing high residual data (Crystal and NMR) that existed previously. But, the idea of combining HDX, cross-linking, MS, and homology modeling to study the structural interactions between protein and dsDNA was innovative and well explained. The authors also expanded upon previous high residual structures of the FOXO4-DBD to come up with structural information for 82-207 AA. The paper is well written and will interest readers in the fields of transcription, biophysics, and structural biology. I would recommend publishing after addressing these minor comments.

Comments:

The authors mentioned that this was the first instance of using HDX MS to study protein and duplex DNA interactions (Line 297). This sentence is misleading because some other groups have already reported studying protein-dsDNA interactions with HDX MS. Recommended to change this sentence. Will this approach always be used to fill in the blanks from higher residual approaches or can independently solve the structure upon changing the parameters like mutation on the interacting residues vs DNA sequences? Methods for the PAGE is missing. What percentage are those native and urea gels are? Why FOXO4-dsDBE-tPt band on urea gel is almost invisible, and the same sample in SDS-PAGE has a second shift? Figure captions are not more descriptive; for example, what are those vertical shaded areas, in figure 3 (page 9)? There are 2 figure 3s in this manuscript. Figure # 3 on page 14 should be figure 5, also look at the figure citations.

Author Response

Rev 4.

The authors presented MS-based approaches for the structural characterization of dsDNA-protein complexes; in this case, the transcription factor FOXO4 and DBE. In general, this article seems to be geared toward creating a new method to evaluate the structural insights towards the binding mechanism of flexible proteins. It is obvious from the manuscript that most of the data generated was used to support existing high residual data (Crystal and NMR) that existed previously. But, the idea of combining HDX, cross-linking, MS, and homology modeling to study the structural interactions between protein and dsDNA was innovative and well explained. The authors also expanded upon previous high residual structures of the FOXO4-DBD to come up with structural information for 82-207 AA. The paper is well written and will interest readers in the fields of transcription, biophysics, and structural biology. I would recommend publishing after addressing these minor comments.

The authors mentioned that this was the first instance of using HDX MS to study protein and duplex DNA interactions (Line 297). This sentence is misleading because some other groups have already reported studying protein-dsDNA interactions with HDX MS. Recommended to change this sentence.

We agree with this point. Appropriate changes have been made.

Will this approach always be used to fill in the blanks from higher residual approaches or can independently solve the structure upon changing the parameters like mutation on the interacting residues vs DNA sequences?

 We strongly believe, that that this approach would be capable to independently solve structure. However, sequentially/structurally related template would be probably necessary.

Methods for the PAGE is missing. What percentage are those native and urea gels are?

PAGE gels appear only in supporting information. PAGE is a well-established method and we found it unimportant to mention this method in detail. The percentage of gels (12%) was incorporated into the caption of Fig. S1.

Why FOXO4-dsDBE-tPt band on urea gel is almost invisible, and the same sample in SDS-PAGE has a second shift?

The band of FOXO4-dsDBE-tPt is very smeared, which corresponds with the low intensity.

Figure captions are not more descriptive; for example, what are those vertical shaded areas, in figure 3 (page 9)?

We agree with this point. Appropriate changes have been made.

There are 2 figure 3s in this manuscript. Figure # 3 on page 14 should be figure 5, also look at the figure citations.

We agree with this point. Appropriate changes have been made.

Round 2

Reviewer 2 Report

The authors have addressed some of my earlier comments adequately, however, the major issue (i.e. the focus of the manuscript and the exaggeration of their main conclusions) has not been resolved.

Author Response

According the reviewer comment, the manuscript was corrected. A part of the conclusion session was deleted since these paragraphs probably contained exaggerated comments not fully supported by experimental data. Also, the focus of the manuscript is currently more  driven toward methodology development.